# On Andean Long-Horned Caddisfly *Brachysetodes* Schmid, 1955 (Trichoptera: Leptoceridae): Discovery of a New Species, Distribution, and an Identification Key

**DOI:** 10.3390/insects16080832

**Published:** 2025-08-12

**Authors:** Gleison R. Desidério, Vitória Santana, Neusa Hamada, Diego G. Pádua, Rodrigo O. Araujo, Andrés Moreira-Muñoz, Pitágoras C. Bispo

**Affiliations:** 1Laboratório de Biologia Aquática (LABIA), Programa de Pós-Graduação em Biociências—Interunidades, Faculdade de Ciências e Letras de Assis (FCLAs), Universidade Estadual Paulista “Júlio de Mesquita Filho” (UNESP), Assis 19806-900, Brazil; pitagoras.bispo@unesp.br; 2Programa de Pós-Graduação em Biodiversidade Neotropical (PPGBN), Universidade Federal da Integração Latino-Americana (UNILA), Foz do Iguaçu 85870-650, Brazil; 3Laboratório de Citotaxonomia e Insetos Aquáticos (LACIA), Programa de Pós-Graduação em Entomologia (PPGEnto), Instituto Nacional de Pesquisas da Amazônia (INPA), Manaus 69067-375, Brazil; vsantana.bio@gmail.com (V.S.); neusaha@gmail.com (N.H.); 4Laboratorio de Entomología General y Aplicada, Centro de Investigación de Estudios Avanzados del Maule, Universidad Católica del Maule, Talca 3460000, Chile; paduadg@gmail.com; 5Instituto de Geografía, Pontificia Universidad Católica de Valparaíso, Valparaíso 2340025, Chile; andres.moreira@pucv.cl

**Keywords:** aquatic insects, Leptocerinae, taxonomy, morphology, biogeography, Chilean Andes, Parque Natural Tricahue

## Abstract

Caddisflies are important insects that live in freshwater environments such as rivers and streams, where they help to maintain healthy ecosystems. Some species build small protective cases or nets using natural materials like sand or plant fragments. These insects are also used to monitor water quality, since many species only survive in clean water. In this study, the scientists focused on a small group of long-horned caddisflies that live only in the Andes Mountains of South America, mainly in Chile. More than 40 years following the last major research work on this group, a new species was discovered in a protected area in the Chilean Andes, called Parque Natural Tricahue. The new species was carefully studied, photographed, and compared with similar species. It can be identified by the unique features of the male’s reproductive structures, such as specific shapes and patterns not found in other species. A detailed map of the known species’ locations and a user-friendly identification guide were also created. This discovery shows that, even in well-studied regions, there is still unknown biodiversity waiting to be found. It highlights the need for continued exploration and supports conservation efforts by increasing knowledge about life in these sensitive mountain ecosystems.

## 1. Introduction

Leptoceridae Leach, 1815, commonly known as long-horned caddisflies, are the second-largest family among Trichoptera [1]. Currently, four subfamilies are recognized: Grumichellinae Morse, 1981; Leptocerinae Leach, 1815; Leptorussinae Morse, 1981; and Triplectidinae Ulmer, 1906 [2]. Among the Leptoceridae of the New World, *Brachysetodes* is placed within the Leptocerinae, considering both morphological [3] and molecular [2,4] phylogenies, although not within any specific tribes of this subfamily.

Historically, *Brachysetodes* Schmid, 1955 was established by Schmid [5] to include three Chilean species: the type species *Brachysetodes trifidus* Schmid, 1955; *B*. *bifidus* Schmid, 1955; and *B*. *quadrifidus* Schmid, 1955. Subsequently, Schmid [6,7] described five additional species: *Brachysetodes extensus* Schmid, 1958; *B*. *forcipatus* Schmid, 1964; *B*. *major* Schmid, 1958; *B*. *spinosus* Schmid, 1958; and *B*. *tripartitus* Schmid, 1964. Further contributions by Flint [8,9,10,11] expanded the genus through the description of new species in the genus (*B*. *bifurcatus* Flint, 1983; *B*. *insularis* Flint, 1968; and *B*. *nublensis* Flint, 1969) and the transfer of *B*. *duodecimpunctata* (Navás, 1916) from the genus *Setodes* Rambur, 1842. However, *B*. *duodecimpunctata* and *B*. *insularis* were later removed from *Brachysetodes* by Holzenthal [12,13] and reassigned to new Neotropical genera as *Achoropsyche duodecimpunctata* (Navás, 1916) and *Amphoropsyche insularis* (Flint, 1968), respectively.

The first comprehensive revision of *Brachysetodes* was conducted by Holzenthal [14], who provided a detailed diagnosis of the genus, described the immature stages, and extended its known distribution from Chile into Argentina. Furthermore, a morphological phylogenetic analysis revealed *Brachysetodes* to be polyphyletic, corroborating Morse’s [3] earlier hypothesis. Under this framework, *Brachysetodes* comprises 10 species restricted to Morrone’s [15] Andean region [16], of which eight belong to *Brachysetodes sensu stricto* (Figure 1A,B), while two, *B*. *bifidus* and *B*. *major,* are classified as *incertae sedis* [14].

Although *Brachysetodes* is a member of Leptocerinae that is widely distributed throughout the Andean region, particularly in Chile [16], a new species of the genus has been discovered nearly four decades after the last comprehensive revision. This study, therefore, aims to describe and illustrate this new species of *Brachysetodes s. str*. based on male adults collected in the Chilean Andes. In addition, a key is provided for the identification of males of the nine species of *Brachysetodes s*. *str*., along with brief phylogenetic and biogeographic remarks.

## 2. Materials and Methods

### 2.1. Study Area, Specimen Collection, Preparation, and Observation

Specimens were collected using a Malaise trap [17] in a tributary ravine of the Tricahue stream, located within Parque Natural Tricahue (Figure 1C,D), which lies in the central subregion of the Maule region in the Chilean Andes (Figure 1A,B). Parque Natural Tricahue, located in the mountainous area of the San Clemente commune, Chile, is an important site for biodiversity conservation. Positioned within a nationally recognized biodiversity hotspot, this private park underwent restoration following severe forest degradation in the 1990s. Today, it serves as a crucial refuge for native species, playing a significant role in preserving the regional biodiversity [18]. Specimens were preserved in 80% ethanol and will be deposited in the Museo Nacional de Historia Natural, Santiago, Chile (MNNC); the Coleção de Invertebrados, Instituto Nacional de Pesquisas da Amazônia, Manaus, Brazil (INPA); and the Coleção de Insetos Aquáticos “Prof. Dr. Cláudio Gilberto Froehlich”, Universidade Estadual Paulista, Assis, Brazil (CIASP), as listed in the examined material.

Specimen preparation followed standard methods described in [19], using a hot 10% KOH solution to diaphanize the abdominal tissues. Male genital structures, once clarified, were examined under a Leica DM5500 B compound microscope (Leica Microsystems, Wetzlar, Germany) and stored according to the procedures outlined in Desidério et al. [20].

### 2.2. Illustrations and Map

Photographs of adults were taken using a Leica DMC4500 digital video camera mounted on a Leica M205A stereomicroscope (Leica Microsystems, Wetzlar, Germany), equipped with an LED illumination dome [21]. The Leica Application Suite (LAS) software (version 4.10.0) was used to capture image stacks of each structure at different focal planes, which were then automatically combined into single images with a greater depth of field using the Helicon Focus^®^ Pro stacking software (version 7.6.4), employing the following rendering method and parameters: method B (depth map), radius 18, and smoothing 2.

Male genitalia were photographed with a Leica DFC295 video camera (Leica Microsystems, Wetzlar, Germany) attached to a Leica DM5500 B microscope. Image stacks were captured with the Leica Application Suite (LAS) software (version 4.2.0) and served as templates for the creation of vector illustrations in Adobe Illustrator^®^ (version 29.6), facilitated by a graphic tablet and pen (Intuos CTL4100, Wacom Technology Co., Saitama, Japan). All photographs and illustrations were edited and assembled into plates using Adobe Photoshop^®^.

Distribution maps were generated using the QGIS software (version 3.34.0—Prizren) using the shapefile of Romano [22], based on the Andean biogeographical regionalization proposed by Morrone [15]. The satellite imagery was obtained from Google Earth^®^. Distribution data for all *Brachysetodes* species were compiled from the Global Biodiversity Information Facility database [23], selecting only records for SPECIES “taxonRank”. For species lacking coordinates, locality data were georeferenced in Google Earth^®^ based on information from the original literature.

### 2.3. Morphological Terminology, Description, and Key

Morphological terminology for head setal warts follows Oláh and Johanson [24]. Terminology for male genitalia adheres to Nielsen [25] and Schmid [5], as interpreted for *Brachysetodes* by Holzenthal [14]. Wing venation terminology is based on the Comstock–Needham system, as applied to Trichoptera by Mosely and Kimmins [26]. To ensure consistency and standardization in descriptive taxonomy, the species description was generated using the DELTA editor (Description Language for Taxonomy) software (version 1.02), based on a morphological character matrix. The identification key is adapted from Holzenthal [14] and focuses exclusively on males of *Brachysetodes s*. *str*., emphasizing genital characteristics; all referenced figures correspond to that work, except for those of the new species.

## 3. Results

### 3.1. Species Description


***Brachysetodes tricahue* Desidério, Santana & Hamada, sp. nov.**


urn:lsid:zoobank.org:act:156E3B9E-7153-48E5-BDDC-264FF6AA1C17

(Figure 2 and Figure 3)

**Diagnosis.** *Brachysetodes tricahue* Desidério, Santana & Hamada, sp. nov. closely resembles *B*. *bifurcatus* and *B*. *nublensis*, particularly in the structure of tergum X and the inferior appendages. All three species share a pair of lateral processes on tergum X, each bearing a pair of medium-sized, stout apicolateral setae, as well as an inferior appendage divided into three processes. However, *B*. *tricahue* sp. nov. can be distinguished by the presence of four long, stout setae grouped at mid-length on tergum X, whereas *B*. *bifurcatus* and *B*. *nublensis* each possess only three setae, which are stout only in *B*. *nublensis*. In the new species, the inferior appendage bears three unequal processes, while, in both *B*. *bifurcatus* and *B*. *nublensis*, these processes are subequal. Both *B*. *tricahue* sp. nov. and *B*. *nublensis* have a concave posterior margin for sternum IX, contrasting with the straight margin in *B*. *bifurcatus*. The mesal process of the inferior appendage is bifid in both *B*. *tricahue* sp. nov. and *B*. *bifurcatus*; however, in the former, the digitate lobes are unequal in length, whereas, in the latter, they are equal. Another diagnostic feature of *B*. *tricahue* sp. nov. is the slender, elongate ventral process of the inferior appendage, which surpasses tergum X, with a basomesal margin that is crenulate and lacks projections or lobes. Differences in the phallic apparatus also support species separation: in *B*. *tricahue* sp. nov. and *B*. *bifurcatus*, the parameres are long, reaching the length of the phallic apparatus, whereas, in *B*. *nublensis*, they are very short.

**Description.** *Adult male*: General coloration pale brown (in alcohol) (Figure 2A,D). Head dorsally with seven setal warts; medioantennal setal wart (ma. sw.) large, obovoid; lateroantennal pair (la. sw.) medium-sized, ellipsoid; occipital pair (occ. sw.) very large, obovoid; postgenal pair (pt. sw.) small, posterad to eye, rounded (Figure 2C); in frontal view, frontogenal pair (fg. sw.) very large, obovoid (Figure 2B). Frontal grooves (ft. g.) V-shaped anteromesally, well pronounced; coronal groove (cn. g.) thread-shaped, weakly pronounced on anterior half of head (Figure 2C). Maxillary palp length formula (I < II = III < IV < V) (damaged in holotype); labial segment length formula (I = II < III) (Figure 2B). Forewing, in alcohol, brown, with lighter anal area, without spots; forewing length 3.71–3.73 mm (*n* = 5); forks I and V present, both with short petiole; fork V about 2× longer than fork I; thyridial cell about as long as 2/3 of discoidal cell (Figure 2E). Hindwing with forks I and V, both with long petiole; fork V about long as fork I; *r-m* crossvein present (Figure 2F). Tibial spur formula 1:2:2.

*Male genitalia*: Segment IX, in lateral view, subtriangular, with anterodorsal margin almost straight; anteromesal margin convex, rounded, strongly produced; anteroventral margin straight; posterior margin produced basally, bearing apex truncated (Figure 3A); in dorsal view, anterior margin with shallow, wide V-shaped mesal incision, strongly sclerotized band extending posteriorly, but not touching the posterior margin; posterior margin produced mesally and straight (Figure 3B); in ventral view, anterior margin with wide V-shaped mesal incision; posterior margin slightly concave, bearing a pair of large, rounded setal warts (vt. sw.) (Figure 3D). Preanal appendages (pr. app.) originating dorsomesally, setose, slender, digitate, about 3/4 length of tergum X in lateral and dorsal views (Figure 3A,B); in dorsal view, apex rounded, bearing long, thin setae (Figure 3B). Tergum X, in lateral view, narrowing apically, apex rounded (Figure 3A); in dorsal view, bearing four long, stout setae grouped at mid-length; divided by V-shaped apicomesal incision extending anteriorly about 2/3 length of the segment, originating pair of paired lateral process (lt. pc.) (Figure 3B); each lateral process (lt. pc.) slender, bearing one pair of short, stout subapical setae on mesal margin and one pair of medium-sized, stout apicolateral setae, apex rounded (Figure 3B,C). Inferior appendage (if. ap.), in lateral view, setose, base broad, divided into three processes of unequal length (Figure 3A); dorsal process (ds. pc.), in lateral and dorsal view, hook-like, medium-sized, curved mesad, apex rounded and directed mesad, bearing numerous short, stout dorsal setae (Figure 3A,B); mesal process (ms. pc.), in lateral view, bifid, with digitate lobes of unequal length (Figure 3A); in dorsal view, directed mesad (Figure 3B); ventral process (vt. pc.), in lateral and dorsal view, slender, very long, surpassing tergum X, narrowing apically slightly curved dorsad, apex rounded and directed posteromesad (Figure 3A,C); in ventral view, basomesal margin crenulate, without projections or lobes, covered by numerous long, thin setae (Figure 3D); basal plate (bs. pt.), in lateral view large, trough-like (Figure 3A). Phallic apparatus, in lateral view, tubular, almost straight along its length; phallobase large, shoe-shaped in lateral, ventral margin strongly sclerotized; endothecal membranes striate apicodorsally, apex bifid, with three medium-sized, thin setae; endothecal spines absent; parameres present, paired, slender, strongly sclerotized, as long as phallic apparatus; apex slightly bifid, covered by numerous short, thin setae, directed posteroventrad in lateral view and posterolaterad in dorsal view; phallotremal sclerite (ph. sc.) well-developed, comma-shaped in lateral view and U-shaped in dorsal view (Figure 3E,F).

**Distribution.** CHILE (Región del Maule) (Figure 1).

**Material examined.** *HOLOTYPE* MALE. CHILE: Región del Maule, San Clemente, Parque Natural Tricahue, Sendero Siempreverde, 35.6701944° S, 71.0544167° W, 730 m a.s.l., ii–viii.2023, R.O. Araújo et al. legs., Malaise trap (MNNC). *PARATYPES*. Same data as holotype, except 739 m a.s.l., iii–viii.2023, R.O. Araújo et al. legs., Malaise trap, 3 males (MNNC), 2 males (INPA), 2 males (MZUSP), 2 males (MNRJ), 2 males (CIASP).

**Etymology.** The specific epithet refers to Parque Natural Tricahue, a natural sanctuary located in a transitional zone between the northern and southern forests of Chile. The park harbors ecosystems rich in biodiversity, including numerous species endemic to the Maule region, making it a unique and significant site for nature conservation. The epithet is used in apposition.

### 3.2. Key to Males of Brachysetodes s. str. (Modified from Holzenthal [14])

1.Segment X composed of a single, long process bearing a pair of long, stout terminal setae (see Figure 12A,B) ……………………………..……….……......………. *B*. *spinosus*

-Segment X composed of two or three flat, blade-like processes; lateral pair each bearing at least a single short, stout terminal or subterminal seta (see Figures 3B, 4B, 6B, and 10B) ………………………………………………………………………….…...……. 2

2.Segment X composed of two flat, blade-like processes (see Figures 3B, 4B, and 8B) ...…..………………………………………………………………………………………… 3

-Segment X composed of three processes: a pair of flat, blade-like lateral processes and a mesal process (see Figures 6B, 10B, 14B, and 17B)………………………………………………………………………………………… 9

3.Inferior appendage divided into two major processes and a small, median, triangular process (see Figure 4A) ……………………………………………………….. *B*. *extensus*

-Inferior appendage divided into three major processes ……………….……….…….. 4

4.Anterior margin of sternum IX with deep, narrow V-shaped mesal incision; median process of inferior appendage not bifid (see Figure 8A); parameres very short; endothecal spines present (see Figure 8D) ........................................................…. *B*. *nublensis*

-Anterior margin of sternum IX with shallow, wide V- or U-shaped mesal incision; median process of inferior appendage bifid (see Figure 3A,C); parameres very long and slender; endothecal spines absent (see Figure 3D,F) ………… …….........……... 5

5.Anterior margin of sternum IX with shallow, wide U-shaped mesal incision (see Figure 3C); tergum X bearing three long, thin setae grouped at mid-length; inferior appendage divided into three processes of equal length (see Figure 3A) ……………………………………………………………………………….….. *B*. *bifurcatus*

-Anterior margin of sternum IX with shallow, wide V-shaped mesal incision; tergum X bearing four long, stout setae grouped at mid-length; inferior appendage divided into three processes of unequal length (Figure 3A,B) …….….......… *B*. *tricahue* sp. nov.

6.Segment X with prominent, spatulate mesal process (see Figures 10B and 14B) …...…..………………………………………………………………………………………6

-Segment X with mesal process upturned and bulbous (see Figure 17A,B) or short and membranous (see Figure 6A,B) ……………………………………………………...… 7

7.Inferior appendage with three processes; mesal process of segment X with two pairs of short, stout, straight setae and three pairs of long, stout, hooked setae (see Figure 14A,B) ……………………………………………………………..………..……. *B*. *trifidus*

-Inferior appendage with four processes; mesal process of segment X with three pairs of thin setae (see Figure 10A,B) ……………...……………......………..…… *B*. *quadrifidus*

8.Inferior appendage L-shaped in lateral view (see Figure 6A); mesal projection of segment X short, membranous, bifid (see Figure 6B); parameres unbranched, long, curved (see Figure 6D) ………………………………...………………..…….. *B*. *forcipatus*

-Inferior appendage approximately C-shaped (see Figure 17A); mesal projection of segment X upturned and bulbous (see Figure 17A,B); parameres trifid (see Figure 17D,E) ……………...……………………………………………….………….. *B*. *tripartitus*

## 4. Discussion

The discovery of *Brachysetodes tricahue* sp. nov. highlights the ongoing potential for taxonomic and biogeographic discoveries within *Brachysetodes*, particularly in the Southern Andes, where the genus remains insufficiently sampled. The genus may correspond to a particular biogeographic element described by Moreira-Muñoz [27] for the Chilean flora: the south–temperate element, which comprises taxa confined to regions south of 33° S in Chile and adjacent Argentina, primarily within temperate forest ecosystems (see [28]).

Although *Brachysetodes* is most diverse in Central Chile, where the highest number of species has been described, the majority of distributional records are concentrated in the Subantarctic region (Figure 1B). The new species, described from the central subregion of the Chilean Andes, adds to the endemic richness of this biogeographically complex area [15,29]. However, the available data are too limited to confidently infer patterns of regional species richness or endemism. Figure 1B summarizes known records, but several species are known from a single locality, and sampling efforts have been uneven across regions. Moreover, the symbols on the map may represent either a single specimen or multiple collection events, depending on the availability of data. At present, it is not possible to determine whether species distributions reflect true ecological patterns or simply sampling gaps. Broader and more systematic surveys across a wider range of localities are needed to better understand the distribution and diversity of *Brachysetodes* in Chile.

*Brachysetodes tricahue* sp. nov., clearly assignable to *Brachysetodes sensu stricto* based on the synapomorphies identified in Holzenthal’s [14] phylogenetic analyses—particularly the features of forewing venation and the male structures of tergum X and the inferior appendages—further refines our understanding of the morphological variation within the genus and provides valuable data for future phylogenetic studies. Morphologically, it is most similar to *B*. *bifurcatus* and *B*. *nublensis*, especially in the trilobed structure of the inferior appendages and the lateral processes of tergum X. However, consistent and diagnostic differences in the number and robustness of the setae on tergum X, the shape and proportions of the inferior appendage processes, and the structure of the phallic apparatus support the recognition of *B*. *tricahue* sp. nov. as a distinct taxon. Notably, the unequal length of the bifid lobes of the median process and the elongate, crenulate ventral process are autapomorphic features distinguishing it from all known congeners.

These subtle but consistent differences reinforce the diagnostic value of male genitalia for species-level identification within *Brachysetodes*, a pattern also observed in other Leptocerinae genera, such as *Amphoropsyche* Holzenthal, 1985 (see [13,30]); *Oecetis* McLachlan, 1877 (see [31,32]); and *Triaenodes* McLachlan, 1865 (see [33]). Furthermore, such morphological distinctions may reflect not only species-level divergence but also ecological or behavioral specializations yet to be explored, potentially linked to the diversity of functional traits suspected in caddisflies [1].

The distribution of *Brachysetodes* species across the Andes reflects a complex biogeographic history. Although the genus occurs throughout the Chilean Andes, its presence in the Argentine Andes is limited to only three species (*B*. *extensus*, *B*. *forcipatus*, and *B*. *quadrifidus*) [14]. Notably, *B*. *forcipatus* and *B*. *quadrifidus* exhibit the broadest distributions, occurring across multiple subregions in both Argentina and Chile (Figure 1B). Their wider ranges may suggest ecological generalism or a greater dispersal capacity compared to more range-restricted species. In contrast, *B*. *tricahue* sp. nov. appears to have a narrow distribution, currently known only from Parque Natural Tricahue in the Maule region, part of the Central Andean subregion [15]. This highlights the significance of local endemism and the need for further faunistic surveys in these underexplored montane habitats, employing effective sampling methods such as Malaise traps.

The occurrence of *Brachysetodes* species in the Subantarctic region, particularly in the Valdivian Forest province (sensu Morrone [15]), aligns with the findings of Sganga et al. [34], who identified this province as the most species-rich area for caddisflies in Argentina. Interestingly, the Patagonian subregion lacks any confirmed records of *Brachysetodes*, suggesting either ecological constraints limiting the genus’s range or the significant undersampling of aquatic insect fauna in this region. The records of *B*. *quadrifidus* and *B*. *extensus* occur near the boundary between the Subantarctic and Patagonian subregions, as illustrated in Figure 1B. This raises important questions regarding the biogeographical precision of such boundaries. In reality, the delimitation between these two subregions is not sharply defined and may vary depending on the classification system or ecological criteria adopted [15]. However, this transitional zone corresponds to one of the most pronounced environmental gradients in the world—the Andean Cordillera. This mountain range, which runs along the western edge of southern South America, blocks humid winds from the Pacific Ocean, resulting in intense rainfall on its western (Chilean) slopes and a marked decline in precipitation toward the eastern (Argentinean) side [35]. Such conditions create ecotonal zones where faunal elements from both subregions may coexist. Therefore, the presence of *B*. *quadrifidus* and *B*. *extensus* near this boundary likely reflects environmental gradients rather than strict biogeographic separation. Further studies incorporating fine-scale environmental data and comprehensive faunistic surveys are needed to clarify the distributional limits and subregional affiliations of these species.

From a phylogenetic perspective, *Brachysetodes* remains problematic. It is considered polyphyletic [3,14], and, while its placement within the Leptocerinae is supported by both morphological and molecular data [2,4], the lack of tribal assignment highlights unresolved evolutionary relationships. The description of new species, accompanied by detailed morphological data and an updated key to males of *Brachysetodes sensu stricto*, provides a crucial foundation for future phylogenetic revisions, ideally integrating both adult and larval stages, as well as molecular datasets.

In conclusion, *B. tricahue* sp. nov. contributes to the known diversity of *Brachysetodes* and enhances our understanding of the Trichoptera biogeography in the Southern Andes. This study establishes a valuable baseline for future research incorporating larval stages, ecological data, and molecular analyses to reassess the monophyly of the genus and refine our understanding of leptocerid evolution and diversification in South America.

## Figures and Tables

**Figure 1 insects-16-00832-f001:**
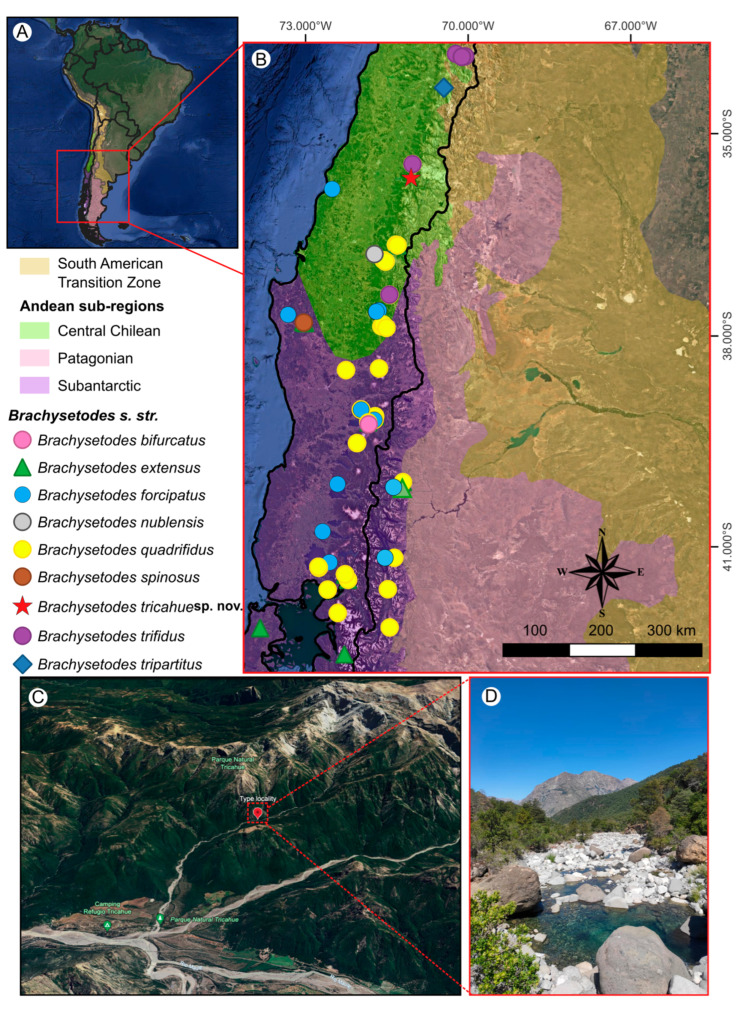
Distribution of *Brachysetodes* species in the Andean region and general views of the collection site of the new species described here: (**A**) map of the Andean region *sensu Morrone* (2015), showing its sub-regions and South American transition zone (highlighted in colors); (**B**) geographic distribution map for all *Brachysetodes* species; (**C**) satellite image from Google Earth^®^ showing the type locality of *Brachysetodes tricahue* Desidério, Santana & Hamada, sp. nov.; (**D**) Parque Natural Tricahue, Región del Maule, Chile—the site where *Brachysetodes tricahue* Desidério, Santana & Hamada, sp. nov. was collected.

**Figure 2 insects-16-00832-f002:**
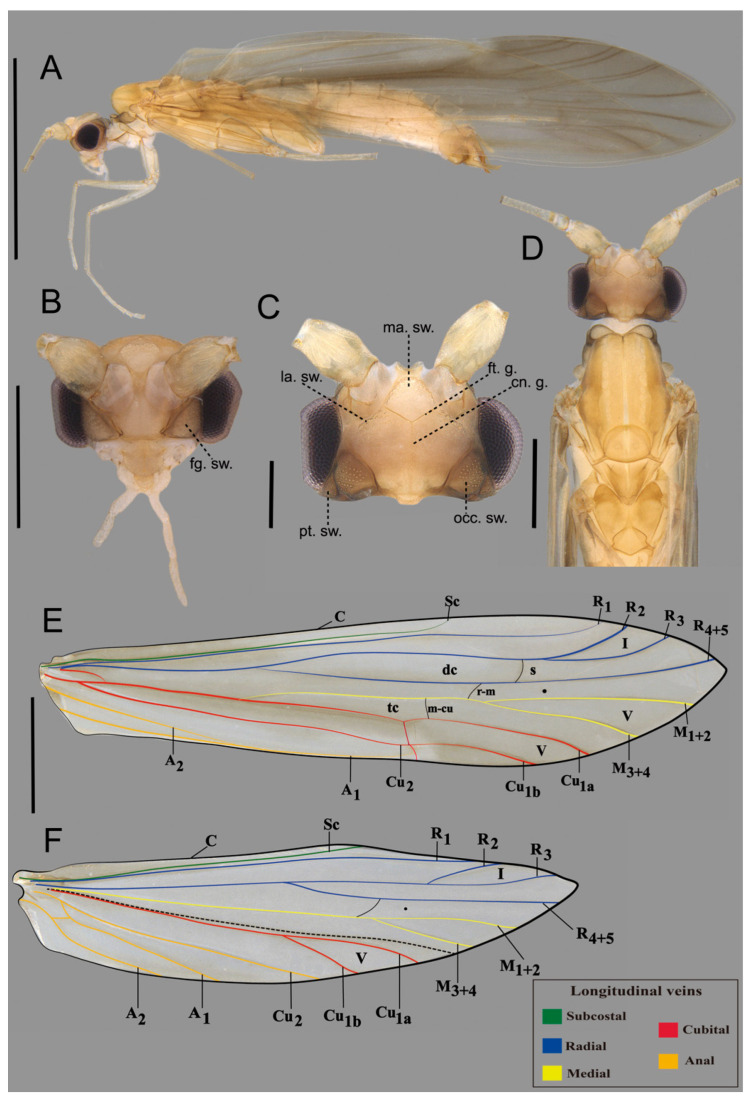
*Brachysetodes tricahue* Desidério, Santana & Hamada, sp. nov., holotype male: (**A**) lateral habitus; (**B**) head, frontal view; (**C**) head, dorsal view; (**D**) head and thorax, dorsal view; (**E**) right forewing, dorsal view (longitudinal veins highlighted in color); (**F**) right hindwing, dorsal view (longitudinal veins highlighted in color). Scale bars in mm: (**A**,**B**,**D**) 0.2; (**C**) 0.05; (**E**,**F**) 0.1.

**Figure 3 insects-16-00832-f003:**
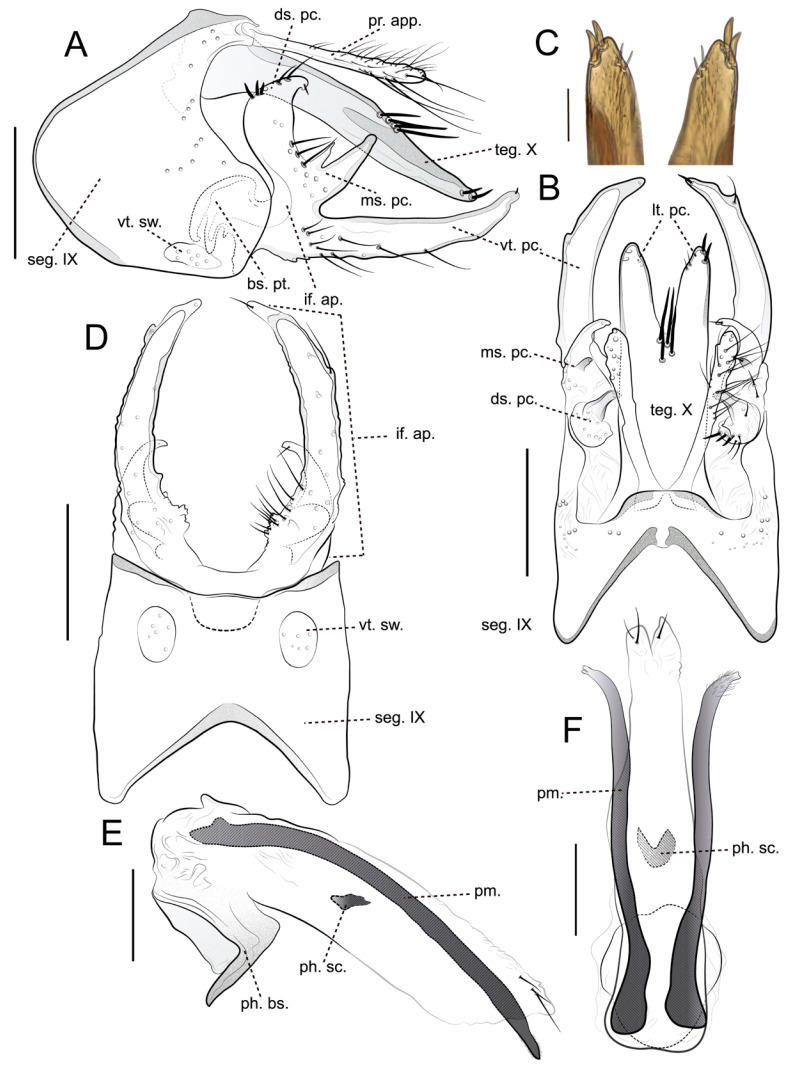
*Brachysetodes tricahue* Desidério, Santana & Hamada, sp. nov., male genitalia (holotype): (**A**) genitalia, left lateral view; (**B**) genitalia, dorsal view; (**C**) apex of lateral process of tergum X, dorsal view (photograph); (**D**) genitalia, ventral view; (**E**) phallic apparatus, left lateral view; (**F**) phallic apparatus, dorsal view. Scale bars in mm: (**A**,**B**,**D**) 0.2; (**C**) 0.05; (**E**,**F**) 0.1.

## Data Availability

All available data are presented in the present study, and specimens are vouchered as indicated in the examined material Section 3.1.

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
