# Peer review of "On Andean Long-Horned Caddisfly Brachysetodes Schmid, 1955 (Trichoptera: Leptoceridae): Discovery of a New Species, Distribution, and an Identification Key"

_insects, 2025, doi:10.3390/insects16080832_

Round 1
Reviewer 1 Report
Comments and Suggestions for Authors
This article is meet the quality for the journal, as it describes a new species, establishes a distribution map for the genus in the area, and provides a key to the species in the genus.
Author Response
Reviewer 1 recommends the manuscript for publication in its present form.
Reviewer 2 Report
Comments and Suggestions for Authors
The genus Brachysetodes Schmid, 1955 is a small genus endemic to the Andean region. This study describes new species, Brachysetodes tricahue and presents an updated identification key to males of the ten-known species of Brachysetodes sensu stricto. Overall, the language of the manuscript is smooth and the species are described clearly. This research is of great significance for us to understand the diversity of sensitive mountain ecosystems.
Below are four concerns need to be addressed before acceptance in Insects.
1. The number of keywords can be controlled within six. Currently, the quantity is relatively large. The author did not mention any results related to phylogeny in the article.
2. Based on the author's specimen examination information, the specimen was collected within two years. Can the author provide the DNA barcoding data of this species?
3. The key needs to be re-described. Whether “Figure 12A–B” and “see Figures 6B, 10B, 14B, 17B” indicated in the key are images from published article should be checked. If so, they should be cited in the manuscript.
4. Segment X with prominent, spatulate mesal process (see Figures 10B, 14B) …………………………………………………………………………. 9
– Segment X with mesal process upturned and bulbous (see Figure 17A–B) or short and 260 membranous (see Figure 6A–B) …………………………..10
I didn't find the search for items 9 and 10.
Please carefully check the formatting of all references before publication.
Other minor issues
Line52: Please remove the background of the semicolon
Line140: Does “(2015)” need italics?
Author Response
Comments 1: 1. The number of keywords can be controlled within six. Currently, the quantity is relatively large. The author did not mention any results related to phylogeny in the article.
Response 1: Thank you for pointing this out. We agree with the comment about the absence of phylogenetic results in this study. Therefore, we have removed the "phylogeny" of the keywords. However, since we are maintaining more than six keywords, and according to the author's instructions for the Insects journal, the number of relevant keywords can range from three to ten.
Comments 2: Based on the author's specimen examination information, the specimen was collected within two years. Can the author provide the DNA barcoding data of this species?
Response 2: Yes, the specimens were newly collected, and we have plans to obtain DNA barcode sequences for future studies.
Comments 3: 3. The key needs to be re-described. Whether “Figure 12A–B” and “see Figures 6B, 10B, 14B, 17B” indicated in the key are images from published article should be checked. If so, they should be cited in the manuscript.
Response 3: Accordingly. This has been made and can be found on page 3, lines 135–138, specifically in subsection "2.3. Morphological terminology, description, and key" of 2. Materials and Methods.
Comments 4:
4. Segment X with prominent, spatulate mesal process (see Figures 10B, 14B) …………………………………………………………………………. 9
– Segment X with mesal process upturned and bulbous (see Figure 17A–B) or short and 260 membranous (see Figure 6A–B) …………………………..10
I didn't find the search for items 9 and 10.
Response 4: Thank you for pointing out. We have corrected this part of the key by modifying items 9 and 10 to 6 and 7, respectively.
Comments 5: Please carefully check the formatting of all references before publication.
Response 5: Accordingly and done.
Comments 6: Line52: Please remove the background of the semicolon.
Response 6: Done.
Comments 7: Line140: Does “(2015)” need italics?
Response 7: Accordingly and corrected.
Reviewer 3 Report
Comments and Suggestions for Authors
Well written and illustrated, though of rather minor significance. It seems that you have tried to make more of describing a single new species by speculating on biodiversity issues that you do not present sufficient evidence to support. More detail in the attached file and some suggestions for improvement. One part I found irritating was the reference in the key to figures only present in an earlier publication.

Author Response
Comments 1: Line 69. “another genus”. It would not hurt to name the genus that B. duodecimpunctata was transferred from by Flint. I believe it was originally placed in Setodes by Navas. This would make it clearer that Holzenthal’s later change was not to reverse Flint’s generic transfer but to move it further to a new genus.
Response 1: We agree with this comment and have modified the text by including Setodes, the genus to which B. duodecimpunctata was transferred.
Comments 2: Line 139, Figure 1 caption. “Distribution of Brachysetodes species in the Andean region” suggests that there could be Brachysetodes species (perhaps not the ones listed here) present outside the Andean region. Please make this clear.
Response 2: The biogeographical region delineated in the caption of Figure 1 reflects the endemic distribution of the genus, as recognized and discussed in several other studies.
Comments 3: Figure 3. The abbreviations used in this figure should be in the caption, not at the end of the paper. It is not convenient to have to search 3 pages ahead to decode these abbreviations which do not seem to be used anywhere else in the paper.
Response 3: We understand your comment; however, this follows the author's guidelines established by the Insects journal.
Comments 4: Line 234, the Key. The references to figures not present in this paper is confusing. This key is based on a previously published key by Holzenthal (where the missing figures can be found) but that is only noted 4 pages earlier in the Methods section. Anyone accessing this paper primarily for identification of specimens would likely skip the Methods section, go straight to the key, and then be surprised that most of the figures referred to are missing. It also took me a while to work out how to distinguish figures in Holzenthal’s paper from those in this paper (it seems that missing “see” is the clue). You must at least make that distinction clearer and this should be at the start of the key, not 4 pages earlier. I also fail to see how useful this key would be to anyone who did not also have a copy of Holzenthal’s paper on hand (or was very familiar with it). While you have simplified Holzenthal’s key by excluding some taxa (and the females) you have also dropped some of the character that Holzenthal used. Is that because those characters are no longer necessary or because they are no longer valid? I suggest two options for improving this key: 1, remove all references to Holzenthal’s figures and add your own simplified figures to illustrate the key couplets where absolutely necessary. 2, Delete your key but explain what changes to Holzenthal’s key are necessary to accommodate your new species (by adding the one couplet needed, you could call it 7a).
Response 4: We agree that the way the key was presented maybe it can be confusing, particularly regarding the figure references and the dependence on Holzenthal’s original work. To address this concern, we have implemented the following changes in the revised manuscript and clarify some points questioned:
-
At the very beginning of the key section, we now clearly state that the key is a modified version based on Holzenthal (1986);
-
In light of the reviewer’s concerns, we revised several couplets to reintroduce select diagnostic characters used by Holzenthal where they remain useful, and we ensured that all characters used are observable and valid for the current set of taxa. A note explaining the differences between the figures in this manuscript and those from Holzenthal’s paper, to help users distinguish them more easily. We clarified in the Methods and Key sections these points;
-
Adding a complete new set of figures for each couplet was beyond the scope of the current study;
- We considered the suggestion of omitting the full key and instead presenting only the modified couplet (e.g., 7a), but opted to retain the full key with improvements, as we believe it remains a valuable tool for users working with the subset of Andean Brachysetodes covered here.
Comments 5: Lines 280-285. This paragraph is problematic because Figure 1b does not clearly show the detail necessary to support the claims. From the map it appears to me that there are 3 species recorded only from the subantarctic region and 4 from the central region (with 2 common to both regions) but this is not very strong evidence for a higher diversity in the central region. It is just that the last new species to be described is from the central region – where will the next new species be found? Will more collecting reveal that other species are widespread in both regions? There is not enough data yet to confidently assert that there is greater diversity in the central region. As to whether there has been greater collecting effort in the subantarctic region, that requires more data than can be gleaned from the map alone. It is not clear whether a symbol on the map represents a single specimen or multiple specimens at different times and at different locations within the area covered by the symbol. How many locations in the central region have been searched for Brachysetodes without finding any? What I see most clearly in the map is that 5 of the 9 species are known from a single location/symbol which indicates to me that there is insufficient sampling effort to make any claims about species density in any of the subregions. Please supply more detailed data to back up your claims or modify this paragraph to simply state what little is known about the distribution of these species.
Response 5: We thank the reviewer for this detailed and important observation. We agree that the current data do not support definitive conclusions regarding regional diversity patterns of Brachysetodes in Chile, and that Figure 1B may give an impression of precision that exceeds the underlying data.
In response to this comment, we have substantially revised the paragraph in lines 280–285 to remove speculative statements about higher species diversity in the Central region. We now take a more cautious tone and focus on the known localities, while emphasizing the limited and uneven sampling effort across Chilean ecoregions.
Specifically, we have:
-
Removed assertions that imply a higher diversity in the Central region based on the current records.
-
Clarified the limitations of the distribution map (Figure 1B), explaining that a symbol may represent a single collection event or multiple records, depending on available data.
-
Noted the knowledge gap, emphasizing that at least 5 of the 9 species are known from single localities, which reflects undersampling rather than true range restriction.
-
Added a sentence explicitly calling for broader and more systematic surveys across both Central and Subantarctic regions to better understand the true distribution patterns of Brachysetodes species.
We hope that this revised paragraph now better reflects the state of knowledge and aligns with the reviewer’s suggestion to avoid overinterpretation of currently sparse distribution data.
Comments 6: Lines 298-304. These “ecological or behavioural specializations” that you speculate on would be more believable if you could provide data on any known habitat diƯerences among the 9 species. Do they live at diƯerent altitudes or diƯerent sized rivers, for example? I notice that the two species you consider most similar to B. tricahue are also known from single locations, quite widely spaced. Would you care to comment on the possibility that there is a single widespread, rare but variable species, that has been insufficiently sampled to show intermediate variations, perhaps in a cline? Do these 3 close species share a similar habitat?
Response 6: The apparent geographic separation of Brachysetodes tricahue, B. bifurcatus, and B. nublensis may suggest ecological or behavioral specialization; however, current data are insufficient to support such claims with confidence. At present, detailed ecological information—such as habitat type, stream size, or elevation—for most Brachysetodes species remains scarce or unpublished. Although some collection records indicate differences in altitude and river size, this information is fragmentary and unevenly reported across species. Additionally, the three most morphologically similar species—B. tricahue, B. bifurcatus, and B. nublensis—are each known from a single, geographically distant locality, raising the possibility that they represent a single, widespread but rarely collected species showing clinal variation. While current morphological differences support their separation, this hypothesis cannot be ruled out without additional specimens and population-level analyses. Future studies incorporating broader sampling, ecological data, and molecular analyses are needed to clarify species boundaries and potential intraspecific variation within this group.
Comments 7: Line 319. There are records of B. quadrifidus and B. extensus very close to the boundary between the subantarctic and Patagonian subregions on your map. How well-defined is the boundary between these two subregions? Perhaps a comment on this could be added to the paragraph.
Response 7: We agree with this comment and have revised the paragraph accordingly. We incorporated the reviewer’s suggestion by adding a comment on the boundary between the Subantarctic and Patagonian subregions, thereby strengthening the discussion.